# Age and gender disparities in administration of opioid for cardiac chest pain in the emergency department

Emad Awad[1]*, Heba Abushanab[2], Raed Darwish[3], Taryn Hunt-Smith[4], Mohamed Shubair[5], Jeffrey Druck[1]

1 Department of Emergency Medicine, University of Utah, Salt Lake City, Utah, United States of America, 2 Department of Instructional Technology and Learning Science, Utah State University, Utah, United States of America, 3 College of Medicine, Ain Shams University, Cairo, Egypt, 4 School of Medicine, University of Utah, Salt Lake City, Utah, United States of America, 5 School of Medicine, University of British Columbia, Vancouver, British Columbia, Canada

* emad.awad@utah.edu

## Abstract

### Background

Previous data have shown sex differences in pain management for patients with cardiac chest pain in the emergency department (ED); however, the joint effect of sex and age on opioid administration has not been well studied. This study aimed to evaluate the combined effect of age and sex on the administration of opioid analgesics, specifically morphine and fentanyl, in ED patients presenting with cardiac chest pain.

### Methods

This retrospective observational study included adults aged 18 years and older who presented to a single tertiary academic ED with acute cardiac chest pain between 2021 and 2025. Patients were categorized into four age–sex groups: older women (>57 years), younger women (18–57 years), older men (>57 years), and younger men (18–57 years). The primary outcome was the administration of intravenous (IV) morphine or fentanyl during the ED visit. Multivariable logistic regression was used to examine the association between these groups and the administration of opioids.

### Findings

Among 1,870 eligible patients, 474 (25.4%) were older women, 323 (17.3%) were younger women, 659 (35.2%) were older men, and 414 (22.1%) were younger men. Compared to older women, all other age–sex groups had higher odds of receiving IV morphine. Younger men had the highest odds (OR 2.19; 95% CI: 1.58–3.04; p < 0.001), followed by older men (OR 1.99; 95% CI: 1.22–3.26; p = 0.006) and younger women (OR 1.48; 95% CI: 0.87–2.52; p = 0.15), although the latter was not

**Data availability statement:** The data used in this study are owned by the University of Utah and were obtained from the University of Utah Emergency Department electronic health record system. Data cannot be shared publicly due to institutional privacy and HIPAA restrictions. Qualified researchers may request access to the data through the University of Utah Institutional Review Board (IRB) or the Office of Quality Compliance by contacting irb@hsc.utah.edu. The authors confirm that they did not have any special access privileges that others would not have and that permission to use the data for this research was granted by the University of Utah.

**Funding:** Faculty Small Grant Program (FSGP) from the Office of the Vice President for Research at the University of Utah. The funder had no role in study design, data collection and analysis, decision to publish, or preparation of the manuscript.

**Competing interests:** The authors declare that no competing interest exist.

statistically significant. IV fentanyl use was low overall and did not differ significantly between groups.

## Conclusions

Older women were significantly less likely to receive IV morphine than men. These findings suggest the need for standardized pain protocols and targeted clinician education to reduce potential bias in ED pain management.

---

## Introduction

Chest pain is one of the most common presentations to emergency departments (EDs) in the United States, accounting for approximately 19.6 visits per 1,000 population annually [1]. It is also a key symptom of acute coronary syndrome (ACS) [1,2]. While chest pain affects both men and women, women are often under-recognized or present with atypical symptoms [3–6] which may lead to differences in how chest pain is reported and managed during ED visits, raising concerns about the quality and equity of care that women receive.

Several studies have documented significant gender disparities in the evaluation and treatment of patients presenting with chest pain in the ED [2,3,7–11]. Women with cardiac chest pain have been reported to experience longer delays from symptom onset to first medical contact and are less likely to receive statins, platelet aggregation inhibitors, or percutaneous coronary intervention (PCI) [12,13]. One study estimated that approximately one-third of the lower PCI utilization in women could be attributed to gender-based disparities [14].

Beyond procedural differences, women are also more likely to experience longer ED wait times and are less likely than men to receive an electrocardiogram (ECG) [8], undergo troponin testing [13], or be prescribed key cardiac medications at discharge, including beta-blockers, lipid-lowering agents, and ACE inhibitors [10]. Similar disparities have also been documented in other cardiovascular emergencies, such as cardiac arrest [15,16].

Until recently, few studies have explored gender differences in pain management for ED patients presenting with chest pain. Addressing this gap, Druck et al. (2025) found that women were significantly less likely than men to receive opioid analgesia. While this study highlighted an important disparity, it left a critical question unanswered: How do age and gender interact to influence opioid administration in this patient population? To date, no studies have specifically examined the combined effect of age and gender on opioid use for chest pain in the ED, representing a significant gap in the literature. Pain in older adults is often undertreated due to challenges in assessment and concerns about opioid safety and polypharmacy. Evidence also suggests that opioid response and dosing may differ by sex and age, with women, particularly older women, receiving lower doses and potentially experiencing compounded disparities in pain management [17,18].

Morphine and fentanyl are two commonly used opioids for managing cardiac chest pain. Current clinical guidelines recommend morphine as a first-line agent for ongoing ischemic pain unresponsive to initial anti-ischemic therapy due to its potent analgesic effects [19,20]. Fentanyl is increasingly used as an alternative, especially in patients with contraindications to morphine, such as hypotension or intolerance [19,21,22].

The main objective of this study was to evaluate the joint association of age and gender with opioid analgesia administration among patients presenting to the ED with acute chest pain. Specifically, we examined the relationship between age-gender groups and the likelihood of receiving opioid analgesics, focusing on intravenous (IV) morphine and IV fentanyl. We hypothesized that older women presenting with acute chest pain would be less likely to receive opioid compared to younger women and men across all age groups.

## Materials and methods

### Study design, setting, and population

This retrospective observational study was conducted at a single ED within a tertiary academic medical center that receives approximately 62,000 emergency visits annually. The study included adult patients (aged 18 years or older) who presented to the ED with acute chest pain between January 2021 and January 2025, and whose chest pain was cardiac in origin. Cardiac origin of chest pain was defined as meeting one or more of the following criteria: (1) elevated cardiac troponin above the institutional upper limit of normal on at least one measurement; (2) ischemic electrocardiographic changes, including ST-segment elevation, ST-segment depression, or T-wave inversion; and/or (3) a final emergency department or hospital discharge diagnosis consistent with acute coronary syndrome (ICD-10 codes I20–I25). Patients with unstable angina, identified through clinical assessment or serial troponin testing without biomarker elevation, were also included. Patients were excluded if they were younger than 18 years, had chest pain not attributable to ACS, or had an alternative diagnosis explaining chest pain despite a positive troponin result (e.g., chest trauma, heart failure, or chronic kidney disease). Additional exclusion criteria included documented allergies to morphine or fentanyl and missing data on key variables such as gender, age, or outcomes. Patients triaged as Emergency Severity Index (ESI) level 1 were also excluded, as these individuals typically required immediate life-saving interventions (e.g., cardiac arrest, decreased level of consciousness, and shock) and were therefore not candidates for analgesia [23]. The study was granted exempt status by the University of Utah IRB as it involved secondary analysis of de-identified, non-interventional data.

### Data source and variables

Data for this study were extracted from the Epic electronic health record (EHR) system using standardized data collection methods. The data were accessed on 17 May 2025. Authors had no access to identifiable participant information during or after data collection.

The primary independent variables were sex and age. To evaluate their interaction, a composite "age–sex" variable was created, stratifying patients into four groups: older women (>57 years), younger women (18–57 years), older men (>57 years), and younger men (18–57 years). The age cutoff of 57 years was selected to align with thresholds used in prior emergency medicine and cardiovascular disparity studies [16,21], ensuring comparability with existing literature rather than equal group sizes. The primary outcome was the administration of IV opioids (morphine or fentanyl) during the ED visit.

Additional clinically relevant variables collected and included in the analysis were: age (years), race/ethnicity, Emergency Severity Index (ESI) triage level, care provider type (Advanced Practice Provider or MD/DO), initial vital signs at triage (systolic blood pressure, heart rate, respiratory rate), door-to-provider time (minutes), emergency department length of stay (hours), and administration of sublingual (SL) nitroglycerin prior to opioid use.

## Statistical methods

Summary statistics for baseline characteristics were calculated for the full cohort and stratified by combined age-gender groups. Continuous variables were presented as means and standard deviations (SD) if normally distributed, or as medians and interquartile ranges (IQR) if not. Normality of continuous variables was assessed using the Shapiro–Wilk test and by visual inspection of histograms and Q–Q plots. Categorical variables were summarized as frequencies and percentages. Associations between baseline characteristics and age-gender groups were assessed using analysis of variance (ANOVA) for normally distributed continuous variables, Kruskal–Wallis tests for non-normally distributed variables, and chi-square tests for categorical variables.

To evaluate the association between age-gender groups and the primary outcome (opioid administration), multivariable logistic regression was performed with older females as the reference group. Models were adjusted for potential confounders identified a priori based on clinical relevance or established associations from the literature. Prior to analysis, regression assumptions, including multicollinearity and linearity of continuous variables with the logit were assessed. Multicollinearity was assessed using variance inflation factors (VIFs), all below 2.0.

As a sensitivity analysis, we repeated the multivariable logistic regression after excluding patients aged 55–59 years to reduce potential misclassification bias between younger and older age groups. This sensitivity analysis was used to assess whether associations in the full cohort held across clearly defined age-gender strata. For all analyses, statistical significance was set at $p < 0.05$. All analyses were conducted using IBM SPSS Statistics version 30, Armonk, NY.

## Results

The original dataset included 2,384 patients who presented to the ED with chest pain. Of these, 319 patients were excluded due to non-cardiac chest pain based on troponin results and final diagnoses. Additionally, 41 patients triaged as Emergency Severity Index (ESI) level 1 were excluded because they were likely in cardiac arrest, critically ill, or had significantly decreased levels of consciousness, making analgesia unlikely. Furthermore, 126 patients with documented allergies to morphine or fentanyl and 28 patients with missing key data were excluded. The final sample comprised 1,870 patients included in the analysis (Fig 1).

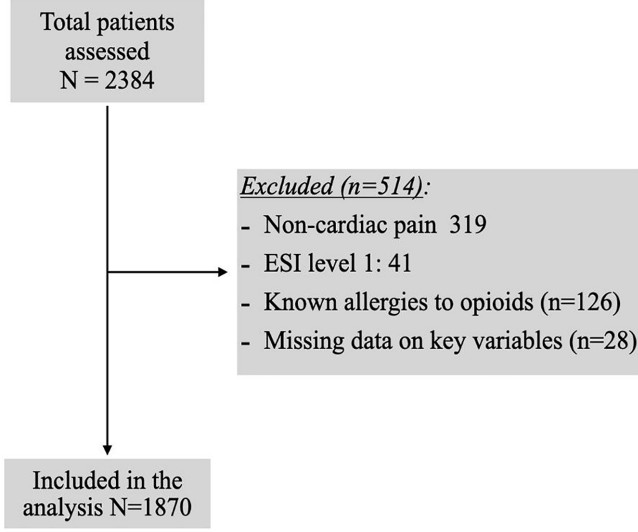

**Fig 1. Study flow chart.**

Of the total cohort (N = 1,870), 474 patients (25.4%) were older women, 323 (17.3%) were younger women, 659 (35.2%) were older men, and 414 (22.1%) were younger men. The majority of patients were White (73.6%), though race distribution differed significantly across groups (p < 0.001). White patients were most prevalent among older men (79.4%) and older women (75.7%). ESI levels also varied significantly (p < 0.001), with older men having the highest proportion triaged as ESI level 2 (51.9%). Significant differences were also observed in initial SBP, HR, DTP times, and ED LOS across the groups. Prior use of SL nitroglycerin differed by group (p < 0.001), with older men more likely to have received it (56.4%) compared to younger women (26.3%) (Table 1).

## Bivariate associations between age-gender groups and administration of IV opioids

Regarding outcome variables, the unadjusted analysis showed that IV morphine administration significantly differed across age-gender groups (p < 0.001). Younger women (18–57 years) received IV morphine most frequently (52.0%), whereas older women (>57 years) received it least frequently (34.2%). In contrast, IV fentanyl use was low overall and did not significantly differ by group (p = 0.35) (Table 1).

## Multivariable associations between age-gender groups and administration of opioids

After adjusting for potential confounders, multivariable logistic regression showed that age–gender group was significantly associated with IV morphine administration. Compared to older women (reference group), younger men (OR 2.19, 95% CI: 1.58–3.04; p < 0.001) and older men (OR 1.99, 95% CI: 1.22–3.26; p = 0.006) had significantly higher odds of receiving IV morphine. Younger women also had higher odds (OR 1.48, 95% CI: 0.87–2.52), although this association was not statistically significant; the confidence interval suggests a potential trend toward lower likelihood of morphine administration compared with older women... Additionally, the analysis showed that prior administration of SL nitroglycerin (OR 0.05, 95% CI: 0.04–0.07; p < 0.001) and IV fentanyl (OR 0.07, 95% CI: 0.04–0.10; p < 0.001) were both associated with lower odds of IV morphine administration (Table 2). Adjusted estimates are shown in Fig 2; full model coefficients are in Table 2.

Regarding IV fentanyl administration, the adjusted analysis showed no statistically significant differences across age–gender groups when compared to older women. Younger women (OR 0.84, 95% CI: 0.38–1.79; p = 0.66) and older men (OR 0.89, 95% CI: 0.43–1.81; p = 0.74) had similar odds of receiving IV fentanyl. Younger men had somewhat higher odds (OR 1.54, 95% CI: 0.97–2.42), although this did not reach statistical significance (p = 0.06). Additionally, prior administration of sublingual nitroglycerin (OR 0.05, 95% CI: 0.03–0.08; p < 0.001) and IV morphine (OR 0.06, 95% CI: 0.04–0.09; p < 0.001) were both associated with significantly lower odds of IV fentanyl use. Finally, patients triaged as ESI level 2 (vs level 3) were significantly less likely to receive IV fentanyl (OR 0.63, 95% CI: 0.43–0.91; p = 0.01). Adjusted estimates are shown in Fig 2; full model coefficients are in Table 3.

## Subgroup analysis

To mitigate potential misclassification bias between younger and older age categories, patients aged 55–59 years were excluded from this analysis. The resulting sample included 1,659 patients: 435 (26.2%) older women, 282 (17.0%) younger women, 600 (36.2%) older men, and 342 (20.6%) younger men. This subgroup analysis yielded results consistent with those of the full cohort. Compared to older women (>59 years), younger men had significantly higher odds of receiving IV morphine (OR 1.89, 95% CI: 1.09–3.27; p = 0.02), as did older men (OR 2.14, 95% CI: 1.54–2.99; p < 0.001). Younger women also had higher odds, but the difference was not statistically significant (Table 4). Adjusted estimates are shown in Fig 3; full model coefficients are in Table 4. Regarding IV fentanyl, the subgroup analysis showed results consistent with the full cohort analysis, revealing no significant differences in IV fentanyl administration across age–gender groups. Adjusted estimates are shown in Fig 2; full model coefficients are in Table 5.

**Table 1. Age-Gender Group Differences in Baseline Characteristics and Outcomes.**

| Variable | Total N = 1,870 | Older Women n = 474 (25.4%) | Younger Women n = 323 (17.3%) | Older Men n = 659 (35.2%) | Younger Men n = 414 (22.1%) | P-value |
|---|---|---|---|---|---|---|
| **Age/years** | | | | | | |
| Mean ± SD | 61.1 ± 15.5 | 72.6 ± 10.1 | 44.0 ± 9.2 | 70.0 ± 9.1 | 46.8 ± 8.2 | <0.001 |
| **Race** | | | | | | |
| White | 1376 (73.6%) | 359 (75.7%) | 211 (65.3%) | 523 (79.4%) | 283 (68.4%) | <0.001 |
| Native American | 72 (3.9%) | 12 (2.5%) | 20 (6.2%) | 17 (2.6%) | 23 (5.6%) | |
| African American | 88 (4.7%) | 19 (4.0%) | 19 (5.9%) | 32 (4.9%) | 18 (4.3%) | |
| Hispanic or Latino | 223 (11.9%) | 62 (13.1%) | 58 (18.0%) | 54 (8.2%) | 49 (11.8%) | |
| Asian | 41 (2.2%) | 8 (1.7%) | 7 (2.2%) | 17 (2.6%) | 9 (2.2%) | |
| Unknown | 70 (3.7%) | 14 (3.0%) | 8 (2.5%) | 16 (2.4%) | 32 (7.7%) | |
| **ESI Level** | | | | | | |
| ESI 2 | 820 (43.9%) | 181 (38.2%) | 108 (33.4%) | 342 (51.9%) | 189 (45.7%) | <0.001 |
| ESI 3 | 1050 (56.1%) | 293 (61.8%) | 215 (66.6%) | 317 (48.1%) | 225 (54.3%) | |
| **Care Provider** | | | | | | |
| ACP | 132 (7.1%) | 31 (6.5%) | 32 (9.9%) | 31 (4.7%) | 38 (9.2%) | 0.006 |
| MD/DO | 1738 (92.9%) | 443 (93.5%) | 291 (90.1%) | 628 (95.3%) | 376 (90.8%) | |
| **SBP** | | | | | | |
| Mean ± SD | 144.1 ± 27.2 | 149.0 ± 26.9 | 137.7 ± 27.3 | 145.1 ± 27.9 | 141.8 ± 25.2 | <0.001 |
| **HR** | | | | | | |
| Mean ± SD | 86.4 ± 20.8 | 83.1 ± 19.6 | 94.1 ± 20.6 | 82.2 ± 19.2 | 91.0 ± 22.1 | <0.001 |
| **RR** | | | | | | |
| Mean ± SD | 18.7 ± 3.8 | 18.5 ± 3.8 | 18.7 ± 3.6 | 18.7 ± 3.8 | 18.9 ± 4.0 | 0.60 |
| **DTP Time/min** | | | | | | |
| Median (IQR) | 14.6 (7.4–32.4) | 15.1 (7.6–34.5) | 16.5 (9.9–44.0) | 13.0 (6.4–24.0) | 15.2 (8.0–31.0) | <0.001 |
| **ED LOS/hr.** | | | | | | |
| Median (IQR) | 6.0 (4.3–8.1) | 6.4 (6.4–8.5) | 6.8 (5.0–9.0) | 5.5 (3.8–7.5) | 5.5 (4.0–7.5) | <0.001 |
| **Prior SL Nitro** | | | | | | |
| No | 992 (53.0%) | 232 (48.9%) | 238 (73.7%) | 287 (43.6%) | 235 (56.8%) | <0.001 |
| Yes | 878 (47.0%) | 242 (51.1%) | 85 (26.3%) | 372 (56.4%) | 179 (43.2%) | |
| **IV Morphine** | | | | | | |
| No | 1051 (56.2%) | 312 (65.8%) | 155 (48.0%) | 373 (56.6%) | 211 (51.0%) | <0.001 |
| Yes | 819 (43.8%) | 162 (34.2%) | 168 (52.0%) | 286 (43.4%) | 203 (49.0%) | |
| **IV Fentanyl** | | | | | | |
| No | 1683 (90.0%) | 421 (88.8%) | 293 (90.7%) | 588 (89.2%) | 381 (92.0%) | 0.35 |
| Yes | 187 (10.0%) | 53 (11.2%) | 30 (9.3%) | 71 (10.8%) | 33 (8.0%) | |

Older women: women >57 years.

Younger women: women 18–57 years.

Older men: men > 57 years.

Younger men: men 18–57 years.

## Discussion

This retrospective analysis examined patients presenting with cardiac chest pain to explore how age and gender jointly influence the administration of opioid analgesics in the ED. The study revealed significant disparities in opioid use based

**Table 2. Association between age–gender groups and administration of IV morphine.**

| Variable | OR | 95% CI | P-value |
|---|---|---|---|
| Age-gender group | | | |
| Women >57 (ref) | – | – | – |
| Women 18–57y | 1.48 | 0.87–2.52 | 0.15 |
| Men > 57y | 1.99 | 1.22–3.26 | 0.006 |
| Men 18–57y | 2.19 | 1.58–3.04 | <0.001 |
| Race | | | |
| White race (ref) | – | – | – |
| Native American | 0.67 | 0.36–1.27 | 0.22 |
| African American | 1.29 | 0.73–2.26 | 0.38 |
| Hispanic or Latino | 1.06 | 0.74–1.53 | 0.74 |
| Asian | 0.75 | 0.33–1.71 | 0.49 |
| Unknown | 1.75 | 0.92–3.33 | 0.09 |
| ESI 2 vs 3 | 1.02 | 0.79–1.30 | 0.90 |
| ACP vs MD/DO | 1.01 | 0.64–1.60 | 0.96 |
| SBP | 1.01 | 0.98–1.01 | 0.96 |
| HR | 1.00 | 0.99–1.00 | 0.95 |
| RR | 0.99 | 0.99–1.00 | 0.17 |
| DTP Time/min | 0.99 | 0.99–1.01 | 0.26 |
| ED LOS/ hr. | 1.00 | 1.00–1.00 | 0.39 |
| Prior SL Nitro | 0.05 | 0.04–0.07 | <0.001 |
| IV Fentanyl | 0.07 | 0.04–0.10 | <0.001 |

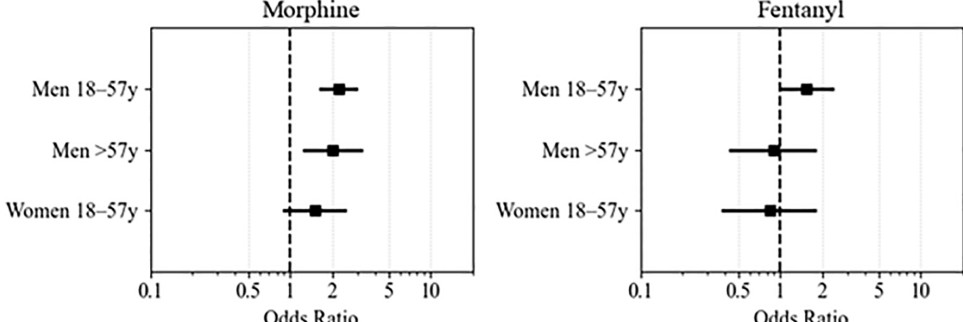

**Fig 2. Forest plots showing adjusted OR for IV opioid administration by age–gender subgroup.** Women > 59 years served as the reference group. Models were adjusted for race/ethnicity, Emergency Severity Index (ESI), provider type (ACP vs MD/DO), systolic blood pressure (SBP), heart rate (HR), respiratory rate (RR), door-to-doctor time, emergency department length of stay (ED LOS), prior sublingual nitroglycerin use, and IV other administration..

on both age and gender, even after adjusting for clinical and demographic factors. Notably, older women were substantially less likely to receive IV morphine compared to younger men and older men.

Our findings align with previous research demonstrating reduced analgesia use among women in EDs adults [24–26]. This is consistent with prior evidence showing that older adults were 14.6% less likely to receive opioids in EDs, even after adjusting for pain severity [27]. Similarly, a recent study involving 15,809 patients across 27,857 hospitalizations found that geriatric patients received significantly fewer opioids compared to younger patients [27]. However, our study extends this literature by quantifying the combined effect of age and gender on opioid administration.

**Table 3. Association between age–gender groups and administration if IV fentanyl.**

| Variable | OR | 95% CI | P-value |
|---|---|---|---|
| Age-gender group | | | |
| Women >57 (ref) | – | – | – |
| Women 18–57y | 0.84 | 0.38–1.79 | 0.66 |
| Men > 57y | 0.89 | 0.43–1.81 | 0.74 |
| Men 18–57y | 1.54 | 0.97–2.42 | 0.06 |
| Race | | | |
| White race (ref) | – | – | – |
| Native American | 0.67 | 0.36–1.27 | 0.22 |
| African American | 1.29 | 0.73–2.26 | 0.38 |
| Hispanic or Latino | 1.06 | 0.74–1.53 | 0.74 |
| Asian | 0.75 | 0.33–1.71 | 0.49 |
| Unknown | 1.75 | 0.92–3.33 | 0.09 |
| ESI 2 vs 3 | 0.63 | 0.43–0.91 | 0.01 |
| ACP vs MD/DO | 1.41 | 0.63–3.15 | 0.41 |
| SBP | 0.99 | 0.98–1.01 | 0.21 |
| HR | 1.00 | 0.99–1.01 | 0.58 |
| RR | 1.01 | 0.96–1.05 | 0.77 |
| DTP Time/min | 0.99 | 0.99–1.00 | 0.46 |
| ED LOS/ hr. | 1.00 | 0.99–1.00 | 0.51 |
| Prior SL Nitro | 0.05 | 0.03–0.08 | <0.001 |
| IV morphine | 0.06 | 0.04–0.09 | <0.001 |

The study also found older women were also less likely to receive morphine compared to younger women, but this difference was not significant. This suggests that within the group of women, age did not affect the likelihood of receiving IV morphine, with both older and younger women having similar rates. Overall, this implies that women, regardless of age, were less likely to receive IV morphine compared to men. Our study also identified lower overall use and distinct prescribing patterns for fentanyl, with no significant differences in its administration observed across the groups. The uniformly low use of fentanyl may reflect institutional opioid stewardship protocols favoring morphine for cardiac pain or concerns about its hemodynamic effects, particularly hypotension or bradycardia, in ischemic presentations.

Another important finding was that patients who received SL nitroglycerin prior to IV morphine administration were significantly less likely to be administered IV morphine compared to those who did not receive nitroglycerin. Similarly, patients who received IV fentanyl were markedly less likely to subsequently receive IV morphine. This inverse association could indicate adherence to protocols designed to avoid polypharmacy or suggest that adequate pain control was achieved with the initial therapy, reducing the need for additional opioids. However, this observation warrants further investigation to determine whether it represents guideline adherence, potential under-treatment of pain, or other factors influencing analgesic administration.

To assess the robustness of our primary findings and minimize potential age misclassification, we conducted a sensitivity analysis by excluding patients aged 55–59 years. This analysis demonstrated that the associations observed in the full cohort persisted within the more clearly defined age–gender strata. These consistent findings suggest that the observed disparities are likely attributable to genuine differences rather than artifacts of arbitrary age cutoffs. Nonetheless, the possibility of residual confounding, such as variations in pain severity or clinician prescribing behaviors, cannot be entirely excluded.

**Table 4. Association between age–gender groups and IV morphine (subgroup).**

| Variable | OR | 95% CI | P-value |
|---|---|---|---|
| Age-gender group | | | |
| Women >59 (ref) | – | – | – |
| Women 18–54y | 1.30 | 0.73–2.33 | 0.37 |
| Men > 59y | 2.14 | 1.54–2.99 | <0.001 |
| Men 18–54y | 1.89 | 1.09–3.27 | 0.02 |
| Race | | | |
| White (ref) | – | – | – |
| Native American | 0.70 | 0.36–1.39 | 0.31 |
| African American | 1.22 | 0.67–2.21 | 0.51 |
| Hispanic or Latino | 1.02 | 0.70–1.49 | 0.91 |
| Asian | 0.76 | 0.33–1.76 | 0.526 |
| Unknown | 1.86 | 0.94–3.68 | 0.074 |
| ESI Level 2 vs 3 | 1.02 | 0.79–1.33 | 0.876 |
| ACP vs MD/DO | 0.93 | 0.58–1.50 | 0.778 |
| SBP (per mmHg) | 1.00 | 0.99–1.01 | 0.978 |
| HR (per bpm) | 1.00 | 0.99–1.00 | 0.135 |
| RR (per bpm) | 0.99 | 0.96–1.03 | 0.682 |
| Door-to-Dr Time (min) | 1.00 | 1.00–1.00 | 0.385 |
| ED LOS/hr. | 0.99 | 0.98–1.01 | 0.401 |
| Prior SL Nitro | 0.06 | 0.04–0.07 | <0.001 |
| IV Fentanyl | 0.08 | 0.05–0.12 | <0.001 |

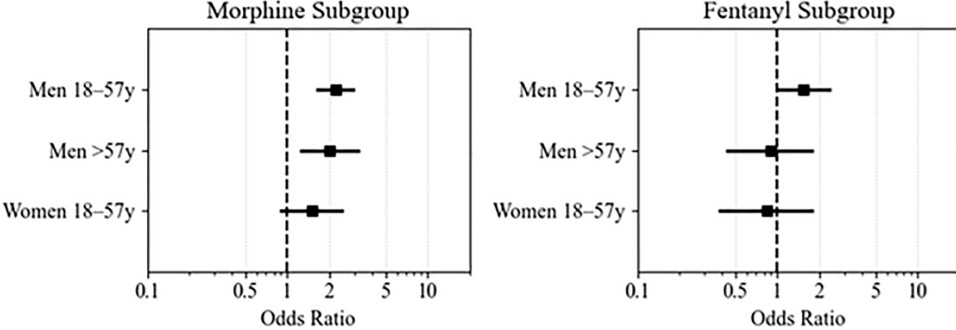

**Fig 3. Forest plots showing adjusted OR for IV opioid administration by age–gender subgroup (subgroup analysis).** *Women > 59 years served as the reference group.* Models were adjusted for race/ethnicity, Emergency Severity Index (ESI), provider type (ACP vs MD/DO), systolic blood pressure (SBP), heart rate (HR), respiratory rate (RR), door-to-doctor time, emergency department length of stay (ED LOS), prior sublingual nitroglycerin use, and other IV opioid administration.

This study confirms significant age- and gender-based disparities in opioid administration for ED patients with acute pain. These findings call for concrete measures to promote equitable care: Standardized pain management protocols, guided by objective clinical criteria, should be implemented to reduce variability in treatment decisions. Concurrently, clinician education initiatives must address implicit biases, particularly in pain assessment for older female patients. Future prospective research should evaluate how these disparities impact pain relief and clinical outcomes. Together, these steps can advance evidence-based, patient-centered pain management while mitigating systemic inequities.

**Table 5. Association between age–gender groups and IV fentanyl (subgroup).**

| Variable | OR | 95% CI | P-value |
|---|---|---|---|
| **Age-gender group** | | | |
| Women >59 (ref) | – | – | – |
| Women 18–54y | 0.89 | 0.37–2.16 | 0.80 |
| Men >59y | 1.77 | 0.97–2.67 | 0.07 |
| Men 18–54y | 0.98 | 0.43–2.22 | 0.95 |
| Race | | | |
| White (ref) | – | – | – |
| Native American | 1.30 | 0.51–3.34 | 0.59 |
| African American | 1.34 | 0.59–3.04 | 0.49 |
| Hispanic or Latino | 1.12 | 0.62–2.00 | 0.71 |
| Asian | 2.63 | 1.03–6.67 | 0.04 |
| Unknown | 0.74 | 0.23–2.37 | 0.62 |
| ESI Level 2 vs 3 | 0.60 | 0.41–0.89 | 0.01 |
| Attendant (ACP vs MD/DO) | 1.02 | 0.47–2.22 | 0.96 |
| SBP | 1.00 | 0.99–1.00 | 0.23 |
| HR | 1.00 | 0.99–1.01 | 0.99 |
| RR | 0.99 | 0.95–1.04 | 0.81 |
| DTP Time/min | 1.00 | 0.99–1.00 | 0.50 |
| ED LOS/hr. | 0.97 | 0.91–1.03 | 0.35 |
| Prior SL Nitroglycerin | 0.05 | 0.03–0.09 | <0.001 |
| IV Morphine | 0.07 | 0.04–0.11 | <0.001 |

While this study provides important insights into age and gender disparities in opioid administration, several limitations should be noted. First, although our findings are likely applicable to other North American hospitals with similar chest pain protocols, they may not generalize to rural or non-academic settings with differing staffing models and pain management practices, or to regions with different practice standards. Second, despite adjusting for key clinical and demographic factors, unmeasured variables such as pain severity scores could influence opioid prescribing patterns. Unfortunately, detailed pain assessments were unavailable in our dataset, limiting our ability to evaluate the appropriateness of treatment decisions. Finally, we did not assess whether the observed differences in opioid administration translated to variations in patient outcomes, such as pain relief or adverse events; this critical question warrants future investigation.

## Conclusions

This study identifies significant age and gender-based disparities in opioid administration for ED patients with cardiac chest pain, with older women were less likely to receive IV morphine than male counterparts of any age. These robust findings, confirmed through sensitivity analyses, advance prior research by demonstrating how intersecting age and gender biases influence acute pain management, particularly in clinical scenarios relying heavily on subjective assessment. To address these inequities, we recommend implementing evidence-based pain protocols using objective criteria and incorporating implicit bias training for clinicians. Building on these findings, critical knowledge gaps must be addressed through targeted investigation. First, prospective studies should examine whether the observed disparities in opioid administration translate to meaningful differences in pain relief or clinical outcomes. Second, qualitative and mixed-methods research is needed to better understand the clinician decision-making processes that contribute to these patterns. Third, intervention studies should evaluate the effectiveness of potential solutions, including standardized pain protocols and implicit bias training programs.

## Acknowledgments

The author thanks the Department of Emergency Medicine at the University of Utah for their support. Special thanks to Joseph Stuppy from the UEMERGE program and Reid Holbrook from the Data Science team for their assistance with data extraction.

## Author contributions

**Conceptualization:** Emad Awad, Raed Darwish, Taryn Hunt-Smith.

**Data curation:** Heba Abushanab, Mohamed Shubair.

**Formal analysis:** Emad Awad, Heba Abushanab, Mohamed Shubair.

**Funding acquisition:** Emad Awad, Jeffrey Druck.

**Investigation:** Emad Awad.

**Methodology:** Emad Awad, Heba Abushanab.

**Project administration:** Emad Awad, Jeffrey Druck.

**Supervision:** Emad Awad, Jeffrey Druck.

**Writing – original draft:** Emad Awad, Heba Abushanab, Raed Darwish, Taryn Hunt-Smith, Mohamed Shubair.

**Writing – review & editing:** Raed Darwish, Taryn Hunt-Smith, Jeffrey Druck.

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
