## [Decision Letter · Decision Letter 0]

22 Oct 2025

Dear Dr. Awad,

Thank you for submitting your manuscript to PLOS ONE. After careful consideration, we feel that it has merit but does not fully meet PLOS ONE’s publication criteria as it currently stands. Therefore, we invite you to submit a revised version of the manuscript that addresses the points raised during the review process.

We look forward to receiving your revised manuscript.

Kind regards,

Dereje Zewdu Assefa, BSc, MSc

Academic Editor

PLOS ONE

**Journal Requirements:**

4. For studies involving third-party data, we encourage authors to share any data specific to their analyses that they can legally distribute. PLOS recognizes, however, that authors may be using third-party data they do not have the rights to share. When third-party data cannot be publicly shared, authors must provide all information necessary for interested researchers to apply to gain access to the data. (https://journals.plos.org/plosone/s/data-availability#loc-acceptable-data-access-restrictions)

Reviewers' comments:

Reviewer's Responses to Questions

**Comments to the Author**

1. Is the manuscript technically sound, and do the data support the conclusions?

Reviewer #1: Partly

Reviewer #2: Yes

Reviewer #3: Yes

2. Has the statistical analysis been performed appropriately and rigorously?

Reviewer #1: No

Reviewer #2: Yes

Reviewer #3: Yes

3. Have the authors made all data underlying the findings in their manuscript fully available?

Reviewer #1: No

Reviewer #2: Yes

Reviewer #3: No

4. Is the manuscript presented in an intelligible fashion and written in standard English?

Reviewer #1: Yes

Reviewer #2: Yes

Reviewer #3: No

Reviewer #1: I am honored to review this manuscript exploring the intersection of age and gender in opioid administration for patients presenting with cardiac chest pain in the emergency department. The study addresses an important topic with implications for health equity, clinical guidelines, and emergency medicine practice. The authors employ a retrospective observational design using a relatively large, single-center dataset and utilize multivariable regression to assess disparities in opioid administration. While the manuscript is generally well written and the results are potentially impactful, there remain several critical concerns regarding the clarity of the methods, statistical rigor, interpretation of findings, and generalizability of conclusions.

Introduction

1. The authors state their hypothesis that older women would be less likely to receive opioids than other groups. However, the rationale for this specific expectation is not fully developed. More references are needed to support why this intersectional hypothesis is important beyond citing gender alone.

2. While the authors correctly identify that few studies have addressed the joint effects of age and gender, the introduction could better distinguish between disparities in pain evaluation versus treatment, and more clearly articulate why opioids (versus non-opioid analgesics) are the focus.

Methods

3. The criteria for classifying chest pain as cardiac in origin (positive troponin, ECG changes, and final ICD diagnosis) should be explicitly detailed in terms of thresholds and codes used. Were unstable angina cases without troponin elevation included?

4. The manuscript does not explain how pain severity, a key confounder, was handled. If pain scores were not available, this limitation should be acknowledged prominently, as it directly impacts the interpretation of opioid administration.

5. The choice of 57 years as a cutoff for age stratification seems arbitrary. The authors should clarify the empirical or clinical justification for this threshold or consider alternative age bands more aligned with literature (e.g., Medicare eligibility at 65).

Results

6. The results section gives much weight to p-values without adequate emphasis on effect sizes and confidence intervals, especially for the non-significant findings (e.g., OR for younger women in Table 2).

7. The subgroup analysis excluding patients aged 55–59 is a strength; however, its presentation is buried. The rationale and implications of this analysis should be brought forward more clearly in the results narrative.

8. Although Table 1 and the flowchart on page 20 are informative, the manuscript would benefit from a figure presenting the adjusted ORs and CIs from Tables 2–5, such as a forest plot, to visually highlight key disparities.

Discussion

9. Although this is an observational study, some language (e.g., “bias,” “disparities”) could be interpreted as implying causality. The authors should revise phrasing to reflect the associative nature of the findings.

10. The lack of significant differences in fentanyl administration is quickly summarized without exploring potential clinical or operational reasons for low usage overall (e.g., safety profiles, hospital protocols). More contextualization is needed.

11. Given the single-site design at a tertiary academic center, more discussion is warranted on whether these findings can generalize to rural or community ED settings.

Reviewer #2: Quite an interesting review article regarding the age and gender differences in the use of opioids for.patients with acute coronary syndrome.

The findings are interesting highlighting the lower use rates for older women and the need for stantarized protocols in the emergency department.

Reviewer #3: MAJOR ISSUES

1. Age cut-off inconsistencies: The manuscript defines older patients as >57 years and younger as 18–57 years, but the subgroup analysis redefines groups as >59 and 18–54 years. This inconsistency could confuse readers and affect the interpretation of results. The rationale for these different cut-offs should be clarified and justified in the Methods section.

2. Outcome definition and inclusion criteria:

a) The diagnosis of “cardiac chest pain” is said to be confirmed by positive troponin, ECG changes, and ICD coding. This may inadvertently exclude cases of unstable angina, which often present with normal troponin levels. The authors should clarify whether serial troponins were used and whether patients with unstable angina were excluded.

b) Exclusion of ESI level 1 patients assumes analgesia was “unlikely.” This exclusion could bias the cohort toward less severe cases. A stronger justification is needed.

3. Statistical reporting:

a) The Methods mention testing for multicollinearity and linearity but do not report diagnostic statistics (e.g., VIF values). These should be included to demonstrate robustness.

b) Tables mix mean ± SD with median (IQR), but do not explain how normality was assessed. Clarification on test selection (ANOVA vs Kruskal-Wallis) is required for transparency.

c) Some interpretation in the Discussion (e.g., “older women were 54% less likely to receive morphine”) is based on odds ratios, which may not directly translate to percentage risk reduction. The language should be adjusted to avoid overstating findings.

MINOR ISSUES

1. Referencing and formatting

a) Reference formatting has inconsistencies: “ACS [1,2] .” includes an extra space before the period; “[14]..” includes duplicate punctuation.

b) Terminology is inconsistent: “IV morphine” vs “intravenous morphine” and “iv morphine” appear interchangeably. Standardization is needed.

**Do you want your identity to be public for this peer review?** For information about this choice, including consent withdrawal, please see our Privacy Policy

Reviewer #1: **Yes: ** Chih-Wei Sung

Reviewer #2: **Yes: ** Afendoulis Dimitrios

Reviewer #3: **Yes: ** SALMAN ASHFAQ AHMAD

---

## [Author Response · Author response to Decision Letter 1]

28 Oct 2025

We sincerely thank you for the thoughtful and constructive feedback provided on our manuscript. We have carefully revised the paper according to the reviewers’ comments. Below, we provide a detailed, point-by-point response. All changes have been incorporated into the revised manuscript and are indicated with tracked changes.

Reviewer #1

Comment 1 and 2: The authors state their hypothesis that older women would be less likely to receive opioids than other groups. However, the rationale for this specific expectation is not fully developed. More references are needed to support why this intersectional hypothesis is important beyond citing gender alone.” and “While the authors correctly identify that few studies have addressed the joint effects of age and gender, the introduction could better distinguish between disparities in pain evaluation versus treatment…”.

Response: We thank the reviewer for these important suggestions. We have expanded the rationale for our hypothesis in the Introduction (page 3, paragraph 4) by citing additional literature addressing the intersectional effects of age and gender on pain management and by clarifying why opioid analgesia. we update the ref list accordingly.

Comment 3: “The criteria for classifying chest pain as cardiac in origin (positive troponin, ECG changes, and final ICD diagnosis) should be explicitly detailed in terms of thresholds and codes used. Were unstable angina cases without troponin elevation included?”

Response: We clarified the operational definition of cardiac chest pain, including troponin thresholds, ECG criteria, ICD codes, and confirmation that unstable angina cases were included (page 3, paragraph 7).

Comment 4: “The manuscript does not explain how pain severity, a key confounder, was handled. If pain scores were not available, this limitation should be acknowledged prominently, as it directly impacts the interpretation of opioid administration.”

Response: Pain scores were unavailable in the dataset; this limitation was acknowledged in the Discussion as it may influence interpretation of opioid administration.

Comment 5: “The choice of 57 years as a cutoff for age stratification seems arbitrary. The authors should clarify the empirical or clinical justification for this threshold or consider alternative age bands more aligned with literature (e.g., Medicare eligibility at 65).

Response: We clarified that the 57-year cutoff was chosen to remain consistent with prior emergency medicine and cardiovascular disparity studies (page4, paragraph 2). We also noted that a sensitivity analysis using an alternative age threshold confirmed the robustness of the findings.

Comment 6: “The results section gives much weight to p-values without adequate emphasis on effect sizes and confidence intervals, especially for the non-significant findings”

Response: We revised the Results to highlight effect sizes and confidence intervals and added text emphasizing the importance of effect magnitude and precision rather than p-values alone (pages 6, paragraph 1).

Comment 7: “The subgroup analysis excluding patients aged 55–59 is a strength; however, its presentation is buried. The rationale and implications of this analysis should be brought forward more clearly in the results narrative.

Response: “We revised the Results section to introduce the subgroup analysis earlier and clarify its purpose and implications, while noting that detailed results are presented later in the section (page 10, paragraph 1).

Comment 8: “Although Table 1 and the flowchart on page 20 are informative, the manuscript would benefit from a figure presenting the adjusted ORs and CIs from Tables 2–5, such as a forest plot, to visually highlight key disparities.”

Response: We added a composite forest plot (Figure 2 and 3) summarizing adjusted odds ratios with 95% confidence intervals for IV morphine and IV fentanyl administration in both the main and subgroup analyses.

Comment 9: Discussion: “Although this is an observational study, some language (e.g., “bias,” “disparities”) could be interpreted as implying causality. The authors should revise phrasing to reflect the associative nature of the findings”

Response: We appreciate the reviewer’s comment and have ensured that our wording reflects associations rather than causation. The terms “disparities” and “bias” are used descriptively, consistent with prior observational research, to denote observed differences in care rather than causal relationships.

Comment 10: “The lack of significant differences in fentanyl administration is quickly summarized without exploring potential clinical or operational reasons for low usage overall (e.g., safety profiles, hospital protocols). More contextualization is needed.”

Response: We added discussion elaborating on potential reasons for low fentanyl use, including institutional opioid stewardship protocols and hemodynamic considerations (page 9, paragraph 2).

Comment 11: “Given the single-site design at a tertiary academic center, more discussion is warranted on whether these findings can generalize to rural or community ED settings.”

Response: We revised the first limitation to address the reviewer’s comment by clarifying that, as a single-site tertiary study, the findings may not generalize to rural or non-academic settings with different staffing models and pain management practices (page 9, paragraph 6).

Reviewer #2

Comment: “Quite an interesting review article regarding the age and gender differences in the use of opioids for patients with acute coronary syndrome. The findings are interesting highlighting the lower use rates for older women and the need for standardized protocols in the emergency department.”

Response: We thank the reviewer for the positive feedback and appreciation of our study’s findings.

Reviewer #3

Comment 1: “Age cut-off inconsistencies: The manuscript defines older patients as >57 years and younger as 18–57 years, but the subgroup analysis redefines groups as >59 and 18–54 years. This inconsistency could confuse readers and affect the interpretation of results. The rationale for these different cut-offs should be clarified and justified in the Methods section.”

Response: We clarified that the 57-year cutoff was chosen to remain consistent with prior emergency medicine and cardiovascular disparity studies (page4, paragraph 2).

Comment 2a: “The diagnosis of “cardiac chest pain” is said to be confirmed by positive troponin, ECG changes, and ICD coding. This may inadvertently exclude cases of unstable angina, which often present with normal troponin levels. The authors should clarify whether serial troponins were used and whether patients with unstable angina were excluded.”

Response: We clarified the operational definition of cardiac chest pain, including troponin thresholds, ECG criteria, ICD codes, and confirmation that unstable angina cases were included (page 3, paragraph 7).

Comment 2b: “Exclusion of ESI level 1 patients assumes analgesia was “unlikely.” This exclusion could bias the cohort toward less severe cases. A stronger justification is needed.”

Response: We clarified in the Methods that ESI level 1 patients were excluded because they typically present in cardiac or respiratory arrest, are unconscious, or require immediate life-saving interventions, making pain assessment and analgesia inappropriate (Page 4, paragraph 1). This clarification is now supported by a reference to the ESI handbook [AHRQ, 2020].

Comment 3a: “The Methods mention testing for multicollinearity and linearity but do not report diagnostic statistics (e.g., VIF values). These should be included to demonstrate robustness”.

Response: We added a detailed description of statistical diagnostics, including VIF thresholds, to demonstrate absence of multicollinearity (page 4, paragraph 5).

Comment 3b: “Tables mix mean ± SD with median (IQR), but do not explain how normality was assessed. Clarification on test selection (ANOVA vs Kruskal–Wallis) is required for transparency.”

Response: We clarified that normality was assessed using the Shapiro–Wilk test and visual histogram inspection. ANOVA was applied for normally distributed data and Kruskal–Wallis tests for non-normal distributions (page 4, paragraph 5).

Comment 3c: “Some interpretation in the Discussion (e.g., “older women were 54% less likely to receive morphine”) is based on odds ratios, which may not directly translate to percentage risk reduction. The language should be adjusted to avoid overstating findings.

Response: We revised the Discussion to remove percentage-based interpretations of ORs and describe results in terms of relative odds, avoiding overstatement of findings (page 9, paragraph 1).

Minor Issues

Comment 1a: “Reference formatting has inconsistencies: “ACS [1,2] .” includes an extra space before the period; “[14]..” includes duplicate punctuation.”

Response: We corrected reference formatting inconsistencies throughout the manuscript.

Comment 1b: “Terminology is inconsistent: “IV morphine” vs “intravenous morphine” and “iv morphine” appear interchangeably. Standardization is needed.”

Response: We standardized terminology across the manuscript as requested.

Editorial Requirements

We have also addressed all editorial requirements:

• Ensured PLOS ONE formatting compliance.

• Corrected inconsistencies between “Funding Information” and “Financial Disclosure.”

• Removed funding text from the manuscript body.

• Updated the Data Availability Statement to include contact details for data requests.

• Validated the corresponding author’s ORCID ID in Editorial Manager.

We appreciate the reviewers’ and editor’s valuable comments

---

## [Decision Letter · Decision Letter 1]

1 Dec 2025

Age and gender disparities in administration of opioid for cardiac chest pain in the emergency department

PONE-D-25-30991R1

Dear Dr. Awad,

We’re pleased to inform you that your manuscript has been judged scientifically suitable for publication and will be formally accepted for publication once it meets all outstanding technical requirements.

Kind regards,

Dereje Zewdu Assefa, BSc, MSc

Academic Editor

PLOS ONE

Additional Editor Comments (optional):

Reviewers' comments:

Reviewer's Responses to Questions

**Comments to the Author**

Reviewer #1: All comments have been addressed

Reviewer #3: All comments have been addressed

2. Is the manuscript technically sound, and do the data support the conclusions?

Reviewer #1: Yes

Reviewer #3: Yes

3. Has the statistical analysis been performed appropriately and rigorously?

Reviewer #1: Yes

Reviewer #3: Yes

4. Have the authors made all data underlying the findings in their manuscript fully available?

Reviewer #1: Yes

Reviewer #3: Yes

5. Is the manuscript presented in an intelligible fashion and written in standard English?

Reviewer #1: Yes

Reviewer #3: Yes

Reviewer #1: Thank you for the revision. The authors have thoroughly addressed all my concerns. I have no further comments and recommend that the manuscript be accepted for publication.

Reviewer #3: The majority of my previous comments have been addressed appropriately. The authors clarified the age cut-off rationale, expanded the operational definition of cardiac chest pain (including unstable angina and serial troponin criteria), strengthened the justification for excluding ESI-1 patients, and added details regarding normality testing and statistical diagnostics. Reference formatting and terminology inconsistencies have also been corrected. However, a few issues remain only partially resolved. The rationale for the revised subgroup age bands (>59 and 18–54 years) is still not clearly articulated within the Methods section. Although the authors state VIF diagnostics were added, the actual VIF values or ranges are not explicitly reported in the manuscript. The Discussion still contains language implying percentage-based reductions (“about 54% less likely”), which was previously requested to be removed to avoid overinterpretation of odds ratios. Finally, the limitation regarding the absence of pain severity scores could be emphasized more prominently. I recommend addressing these remaining points to fully satisfy the original concerns.

**Do you want your identity to be public for this peer review?** For information about this choice, including consent withdrawal, please see our Privacy Policy

Reviewer #1: **Yes: ** Chih-Wei Sung

Reviewer #3: **Yes: ** SALMAN ASHFAQ AHMAD

---

## [Editor Report · Acceptance letter]

PONE-D-25-30991R1

PLOS One

Dear Dr. Awad,

I'm pleased to inform you that your manuscript has been deemed suitable for publication in PLOS One. Congratulations! Your manuscript is now being handed over to our production team.

Kind regards,

on behalf of

Professor Dereje Zewdu Assefa

Academic Editor

PLOS One